# The phonon thermal Hall angle in black phosphorus

Xiaokang Li [1] ✉, Yo Machida [2], Alaska Subedi[3,4], Zengwei Zhu [1] ✉, Liang Li[1] & Kamran Behnia [5] ✉

The origin of phonon thermal Hall Effect (THE) observed in a variety of insulators is yet to be identified. Here, we report on the observation of a thermal Hall conductivity in a non-magnetic elemental insulator, with an amplitude exceeding what has been previously observed. In black phosphorus (BP), the longitudinal ($\kappa_{ii}$), and the transverse, $\kappa_{ij}$, thermal conductivities peak at the same temperature and at this peak temperature, the $\kappa_{ij}/\kappa_{jj}/B$ is $\approx 10^{-4}$–$10^{-3}$ T$^{-1}$. Both these features are shared by other insulators displaying THE, despite an absolute amplitude spreading over three orders of magnitude. The absence of correlation between the thermal Hall angle and the phonon mean-free-path imposes a severe constraint for theoretical scenarios of THE. We show that in BP a longitudinal and a transverse acoustic phonon mode anti-cross, facilitating wave-like transport across modes. The anisotropic charge distribution surrounding atomic bonds can pave the way for coupling between phonons and the magnetic field.

Thermal Hall effect (THE) refers to the emergence of a transverse thermal current by a longitudinal thermal gradient, in presence of a magnetic field. In metals, it is intimately linked to the electrical Hall effect through the Wiedemann-Franz law. Following its original discovery[1], thermal Hall effect was observed in a wide variety of insulating solids[2–10]. Its origin is controversial in exotic cases[11–14], but in strontium titanate[7], a non-magnetic insulator, THE is undoubtedly caused by phonons and is drastically reduced by the introduction of extrinsic atoms[15,16].

These observations motivated numerous theoretical proposals for a thermal Hall signal produced by heat-carrying phonons[17–25]. They can be broadly classified as either intrinsic (invoking the peculiarities of the phonon spectrum) or extrinsic (referring to consequences of specific phonon scattering mechanisms).

What are the minimal ingredients required to produce a detectable THE? To what extent the effect seen in various families share a common origin? Here, we address these two questions by reporting on the observation of thermal Hall effect in black phosphorus (BP), the simplest insulator known to display THE. It is not only non-magnetic, but also lacking ionic bonds. Interestingly, while the amplitude of the thermal Hall conductivity in BP exceeds what was found in all other insulators, the transverse and the longitudinal thermal conductivities peak at the same temperature and their ratio has a similar amplitude. We note that the charge distribution is anisotropic and atomic vibrations can therefore respond to the magnetic field. We show that two out of the three acoustic branches anti-cross and are close to degeneracy in the momentum space, and therefore, energy transfer across harmonic vibrational states[26,27] plays a role in setting the amplitude of thermal conductivity and since the charge distribution is anisotropic, magnetic field can couple to atomic vibrations. The austere context of our experimental observation strongly constrains theoretical scenarios.

## Results

### Thermal Hall effect in black phosphorus

Bulk black phosphorus is a stack of puckered honeycomb layers[28,29] (Fig. 1a). Phosphorus atoms have two distinct sites marked in blue and red. The x- and z- axes correspond to the armchair and zigzag directions of the BP layer plane, following the convention used in ref. [30]. The experimental setup is shown in Fig. 1b (see the method for details).

[1]Wuhan National High Magnetic Field Center and School of Physics, Huazhong University of Science and Technology, Wuhan, China. [2]Department of Physics, Gakushuin University, Tokyo, Japan. [3]Centre de Physique Théorique, École Polytechnique, CNRS, Université Paris-Saclay, Palaiseau, France. [4]Collège de France, Paris, France. [5]Laboratoire de Physique et d'Étude des Matériaux (ESPCI—CNRS—Sorbonne Université), PSL Research University, Paris, France. ✉e-mail: lixiaokang@hust.edu.cn; zengwei.zhu@hust.edu.cn; Kamran.Behnia@espci.fr

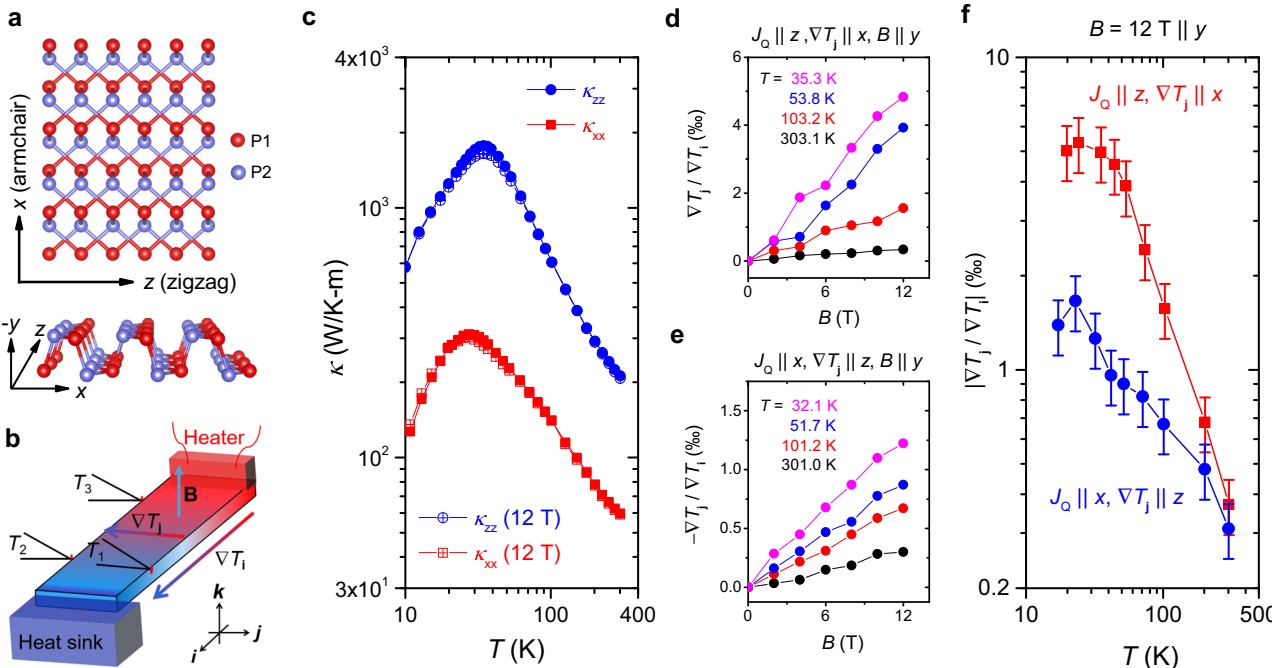

**Fig. 1 | Lattice structure, setup and thermal transport. a** Top (two layers) and side (single layer) view of the lattice of black phosphorus (BP). P atoms, marked in blue and red, belong with atomic sites with distinct environments. The $x$ and $z$ axes respectively correspond to the armchair and zigzag directions of the BP layer plane. **b** The setup for measuring longitudinal and transverse thermal conductivities. **c** Thermal conductivity along the armchair ($\kappa_{xx}$) and the zigzag ($\kappa_{zz}$) orientations.

There is a large in-plane anisotropy, as found by previous studies[31,32]. **d, e** Field dependence of the thermal Hall angle at different temperatures with the heat current $J_Q$ along the $z(x)$-axis, and the transverse temperature gradient along the $x(z)$-axis. In both cases, the magnetic field $B$ is along the $y$-axis. **f** The temperature dependence of the thermal Hall angle for two different configurations at 12 T.

Figure 1c displays the temperature dependence of the longitudinal thermal conductivity. As found in previous studies[31,32], there is a large anisotropy. $\kappa_{xx}$ peaks to 311 WK$^{-1}$m$^{-1}$ at 27 K and $\kappa_{zz}$ to 1770 WK$^{-1}$m$^{-1}$ at 34 K. This is a consequence of the larger sound velocity along the zigzag direction.

The field dependent thermal Hall angle $\nabla T_j / \nabla T_i$, is shown in Fig. 1d, e for two different configurations and at four different temperatures. In both configurations, it is linear in magnetic field. Interestingly, $\nabla T_j / \nabla T_i$ is anisotropic: It attains $5.0 \times 10^{-3}$ at 12 T and 35.3 K for one configuration and $1.2 \times 10^{-3}$ at 12 T and 32.1 K for the other. This fourfold anisotropy is clearly seen in the temperature dependence of $\nabla T_j / \nabla T_i$ (Fig. 1f).

Combining the thermal Hall angle (Fig. 1f), the longitudinal thermal conductivity (Fig. 1c), leads us to the thermal Hall conductivity, shown in Fig. 2a. The inset shows the contribution of electrons to THE can be estimated through the Wiedemann-Franz law. At low temperatures, $\kappa_{ij}^e \approx L_0 \sigma_{ij} T \ll \kappa_{ij}$ and therefore electrons can be totally neglected below 50 K and near the peaks in thermal conductivities. Strikingly, $\kappa_{zx}$ and $-\kappa_{xz}$ become very close to each other. Within experimental margin, they are equal, as expected by the Onsager reciprocal relations for diffusive transport ($\kappa_{ij}(H) = \kappa_{ji}(-H) = -\kappa_{ji}(H)$)[1,33]. The inequality between the two off-diagonal components at high temperature can be tracked to the gradual emergence of a sizeable thermoelectric response at high temperature and therefore, a significant difference between the measured thermal conductivity (in absence of charge current) and the true Onsager coefficient (which is thermal conductivity in absence of the electric field). A similar phenomenon was observed in the case of dilute metallic strontium titanate[16] (see the Supplementary Material for more details).

Figure 2b, c compare the temperature dependence of longitudinal and transverse thermal conductivity. Multiplying $\kappa_{xz}(T)$ and $\kappa_{zx}(T)$ by a factor of $-150$ and 1000 respectively, one finds that $\kappa_{ij}(T)$ and $\kappa_{ii}(T)$ peak almost at the same temperature for both configurations. It's

worth noting that the $\kappa_{ij}/\kappa_{ii}$ ratio is anisotropic. This is unavoidable and due to the combination of the Onsager reciprocity ($\kappa_{ij} = -\kappa_{ji}$) and the anisotropy of longitudinal thermal conductivity ($\kappa_{ii} \neq \kappa_{jj}$).

## Universal features of thermal Hall effect in different insulators

Let us now compare BP with other insulators. The peak amplitude of $\kappa_{ij}$ in BP - 2.2 WK$^{-1}$m$^{-1}$ is four orders of magnitude larger than what was seen in Tb$_3$Ga$_3$O$_{12}$[1], almost two orders of magnitude larger than the values reported for cuprates[5,6] and SrTiO$_3$[7], and almost twice what was recently reported in Cu$_3$TeO$_6$[9]. On the other hand, the thermal Hall angle remains in the same narrow range of 0.1–1% under a magnetic field of about 10 T. As seen in Fig. 2d, across several orders of magnitude variation, longitudinal and transverse thermal conductivity scale with each other. Using, the magnetic length, $\ell_B = \sqrt{\hbar/eB}$ one can extract a length scale, $\lambda_{tha}$, from this angle: $\lambda_{tha}^2/\ell_B^2 = \kappa_{ij}/\kappa_{jj}$. In the case of electrons in the weak-field limit, extracting a length in this way from the Hall angle will give rise to a geometric average of the mean-free-path and the Fermi wavelength. In a similar manner, in some scenarios for phonon Hall response, one expects a correlation between the Hall angle and the mean-free-path.

Intriguingly, for all these solids, $\lambda_{tha}$ remains between 2 and 7 angstroms. Such a length is comparable to the shortest possible phonon wavelength allowed by the interatomic distance. Our observation that it does not correlate with the phonon mean-free-path, which varies by more than three orders of magnitude among these solids, excludes many (but not all) 'extrinsic' scenarios.

A second universal feature is shown in Fig. 3. As first noted in ref. [7], the transverse and longitudinal conductivities occur peak almost the same temperature in all insulators displaying THE. Thus, the thermal Hall response is always maximal when the wave-vector of the heat-carrying acoustic phonons have sufficiently shrunk to make Umklapp scattering irrelevant, but boundary scattering is not yet the dominant scattering mechanisms.

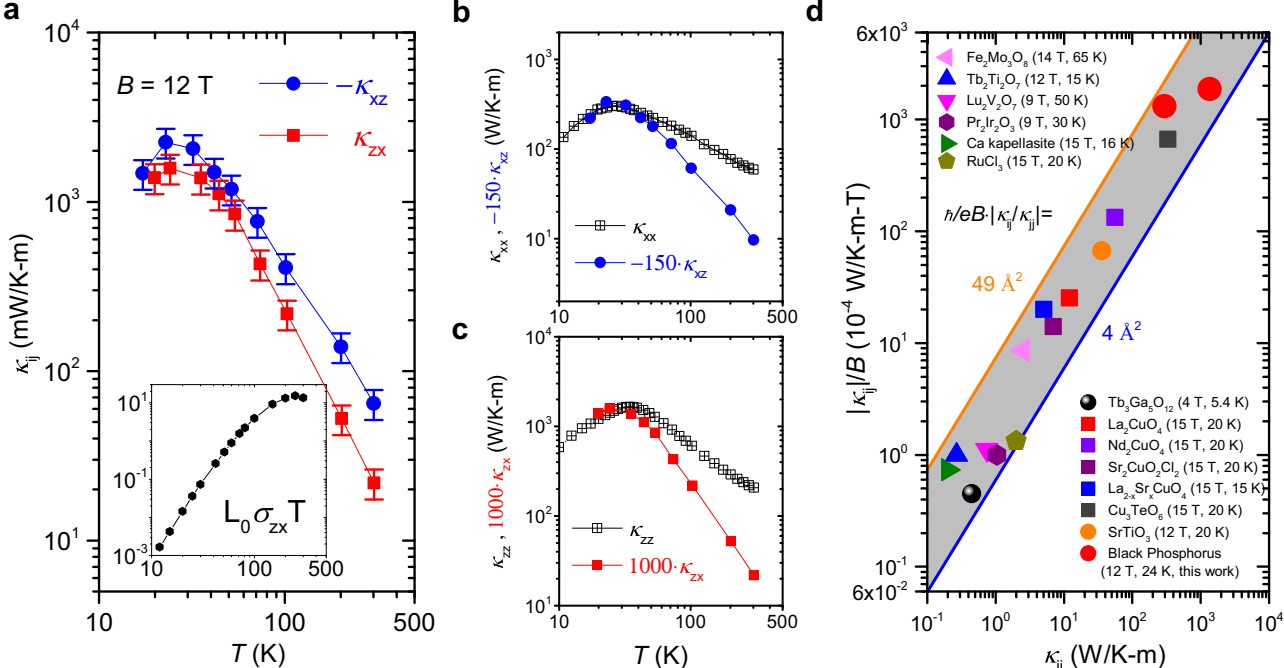

**Fig. 2 | Thermal Hall conductivity. a** Temperature dependence of the measured off-diagonal thermal conductivities, $\kappa_{zx}$ and $-\kappa_{xz}$. At low temperature, they become equal to each other within experimental margin, as expected by Onsager reciprocity. With warming towards room temperature a difference arises, presumably due to non-negligible contribution of the thermoelectric response to transverse heat flow. **b** Comparison of the temperature dependence of $\kappa_{xz}$, multiplied by $-150$ and $\kappa_{xx}$. **c** Comparison of the temperature dependence of $\kappa_{zx}$, multiplied by 1000 and $\kappa_{zz}$.

Note that they all peak almost at the same temperature. **d** Comparison of the transverse $\kappa_{ij}/B$ and the longitudinal $\kappa_{jj}$ thermal conductivity in different insulators (source:[1,3,5–7,9,10,14,45–47]). Even though the longitudinal thermal conductivity $\kappa_{jj}$ varies by 4 orders of magnitude, their ratio $\kappa_{ij}/\kappa_{jj}/B$ remains within the range of $\approx 10^{-4}$–$10^{-3}$ T$^{-1}$. The length scale $\lambda_{tha} = \ell_B \cdot \sqrt{\kappa_{ij}/\kappa_{jj}}$ remains between 2 and 7 Å, equivalent to the shortest phonon wavelength allowed by the distance between atoms.

## Inter-branch diffusion and anisotropic charge distribution

Even in the case of BP, 'extrinsic' scenarios cannot be excluded. Since the inversion center is not at an atomic site, a vacancy breaks the local inversion center and can generate skew scattering or side jump. In the scenario put forward by Guo et al.[25], resonant coupling between phonons and dynamical effects, generates a 'side jump' thermal Hall effect, where the thermal Hall angle does not scale with longitudinal thermal conductivity, in agreement with experiments. However, the suppression of THE by the introduction of extrinsic atoms in strontium titanate[15,16] and the absence of correlation between the thermal Hall angle and the phonon mean-free-path constitute serious challenges for any 'extrinsic' scenario.

Let us now consider those features of BP, which can nourish an 'intrinsic' scenario. The phonon spectrum of BP[34] with a focus on energies below 100 cm$^{-1}$ is shown in Fig. 4a. At 30 K, only the three acoustic modes (one longitudinal and two transverse) are thermally populated. Figure 4b shows the angle dependence of their wave-vector with an energy of $k_B T_{peak} \approx 20$ cm$^{-1}$, which roughly corresponds to the peak temperature. As seen in the figure, two acoustic modes, the longitudinal (LA) and a transverse mode (TA1) display a pronounced anti-crossing and become almost degenerate along the x-axis.

A second relevant feature is the spatial distribution of charge density in bulk BP. Computed and highlighted by Hu et al.[35] (see Fig. 4c), it is highly orientational. We also note that dipole-active phonon modes have been observed in BP by infrared spectroscopy[36], suggesting the presence of unevenly distributed positive and negative charges. Therefore, even in this covalent solid, the magnetic field can couple to phonons through its influence on charge distribution. Interestingly, Valagiannopoulos et al.[37] have found that there is a remarkable anisotropy of electromagnetic propagation in black P. They calculated the real and imaginary parts of the dielectric constant and found that an electromagnetic wave is damped much less along

the 'zigzag' direction than along the 'armchair' direction (see the Supplementary Material for more details).

Note that at peak temperature, the phonon mean-free-path is well below the sample dimensions (see the Supplementary Material for more details), implying a source of thermal resistivity in addition to boundary scattering. Inter-branch coupling between harmonic vibrations was recently invoked to explain the glass-like thermal conductivity of many crystalline solids[26,27]. Our result calls for a theoretical examination of the role of magnetic field in such a context.

## Discussion

In magnetic insulators, collective excitations may couple to phonons and generate a thermal Hall effect[24]. In contrast, in BP, as well as in strontium titanate, phonons are the only identified collective excitations. The two solids share at least two uncommon features. In both, the phonon mean-free-path is not a monotonous function of temperature, which has been tracked to abundance of momentum-conserving phonon-phonon collisions[34,38]. In both, there is a non-trivial coupling between distinct phonon modes[39].

We saw that the transverse and longitudinal conductivities peak at the same temperature in a variety of insulators. This strongly suggests a prominent role played by heat-carrying acoustic phonon in generating the THE even in magnetic materials. However, this does not exclude a role played by magnetism. In magnetic materials, a phonon can couple to another bosonic mode (and possibly a magnon) specific to each material.

In summary, phonons of black phosphorus generate a $\kappa_{ij}$ larger than what was reported in any other insulator. The ratio of $\kappa_{ij}$ to $\kappa_{jj}$ in this system is comparable to other insulators and in all cases, the longitudinal and transverse thermal conductivities peak at the same temperature where the phonon wavelength and the magnetic length are comparable in size. The result shortens list of ingredients required to produce a phonon THE.

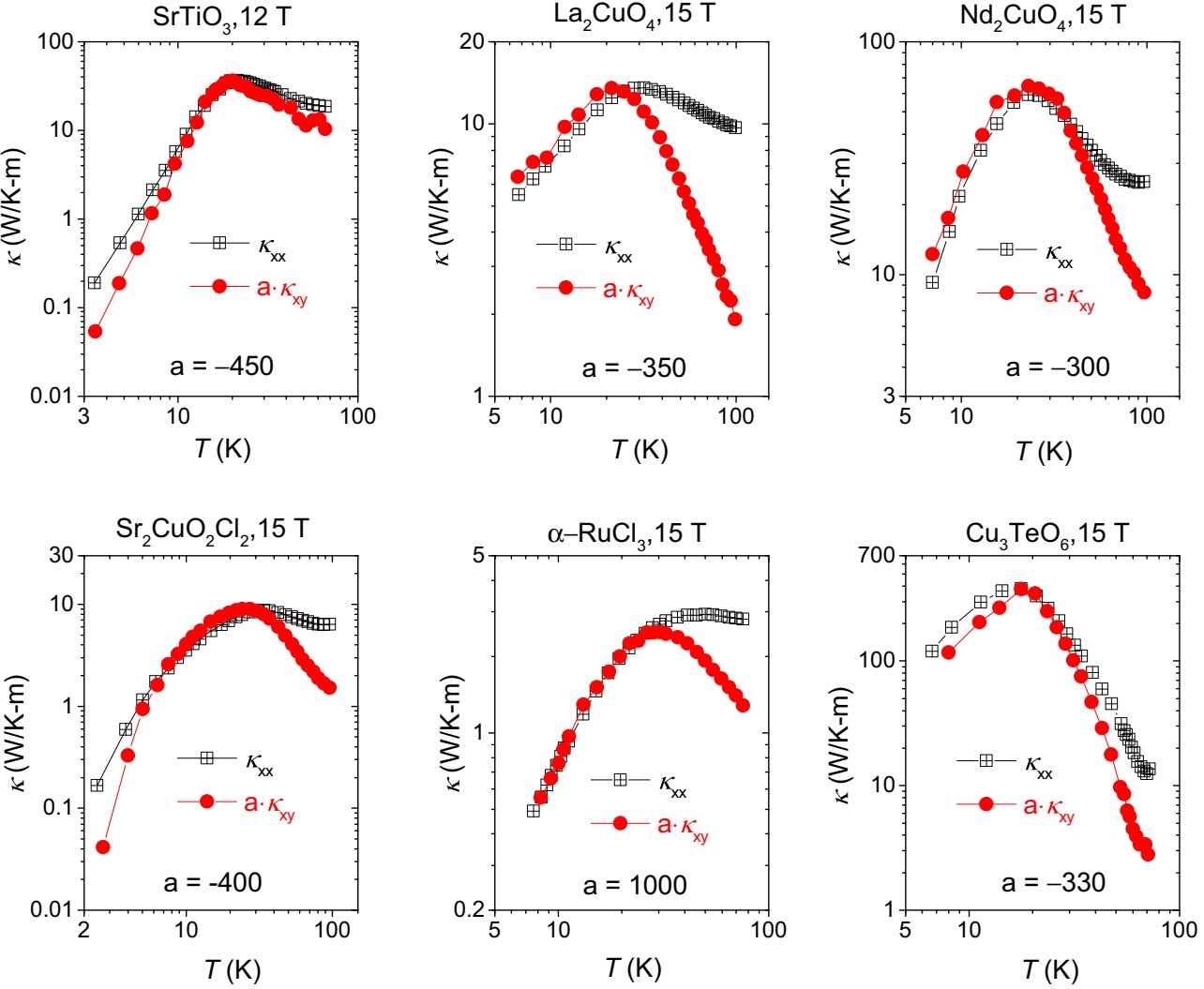

**Fig. 3 | Peaks in $\kappa_{ij}(T)$ and in $\kappa_{ii}(T)$.** Comparison of $\kappa_{ij}(T)$ with $\kappa_{ii}(T)$ in six different insulating materials: $SrTiO_3$[7], $La_2CuO_4$[6], $Nd_2CuO_4$[6], $Sr_2CuO_2Cl_2$[6] $\alpha$-$RuCl_3$[13] and $Cu_3TeO_6$[9]. In each case, $\kappa_{ij}(T)$ is multiplied by an amplification factor $a$ varying from (−)300 to (−)1000.

## Methods

### Samples

Black phosphorus crystals synthesized under high pressure came from two different sources. Samples #1-1, #1-2, #1-3, cut and cleaved from the same mother crystal, were obtained commercially. Samples #2-1, #2-2, also cut and cleaved from the same mother crystal, provided by Prof. Yuichi Akahama (University of Hyogo). Samples #1-1, #2-1 and #2-2 were used for thermal transport measurements, the samples #1-2, #1-3 were used for electrical transport measurements.

### Measurements

All transport experiments were performed in a commercial measurement system (Quantum Design PPMS) within a stable high-vacuum sample chamber. Electrical transport responses were measured by a standard four-probe method using a current source (Keithley6221) and a DC-nanovoltmeter (Keithley2182A). In thermal transport measurements, both one-heater-three-thermocouples (type E) and one-heater-three-thermometers (Cernox 1030) techniques were employed to simultaneously measure the longitudinal and transverse thermal gradient. The thermal gradient in the sample was produced through a 4.7 kΩ chip resistor alimented by a current source (Keithley6221). The DC voltage on the heater and thermocouples (thermometers) was measured through the DC-nanovoltmeter (Keithley2182A). The thermocouples, the heat-sink, and the heater were connected to samples directly or by gold wires with a 50 microns diameter. All contacts on the sample were made using silver paste. Error bars in the main figures represent one standard deviation.

The longitudinal ($\nabla T_i = (T_3 - T_2)/l$) and the transverse ($\nabla T_j = (T_1 - T_2)/w$) thermal gradient generated by a longitudinal thermal current $J_Q$ were measured. They lead to the longitudinal ($\kappa_{ii}$) and the transverse ($\kappa_{ij}$) thermal conductivity, as well as the thermal Hall angle ($\nabla T_j / \nabla T_i$):

$$\kappa_{ii} = \frac{Q_i}{\nabla T_i} \qquad (1)$$

$$\frac{\nabla T_j}{\nabla T_i} = \frac{\kappa_{ij}}{\kappa_{jj}} \qquad (2)$$

$$\kappa_{ij} = \frac{\nabla T_j}{\nabla T_i} \cdot \kappa_{jj} \qquad (3)$$

Here $l$, $w$, $Q$ are the distance between longitudinal thermocouples, the sample width and the heat power respectively.

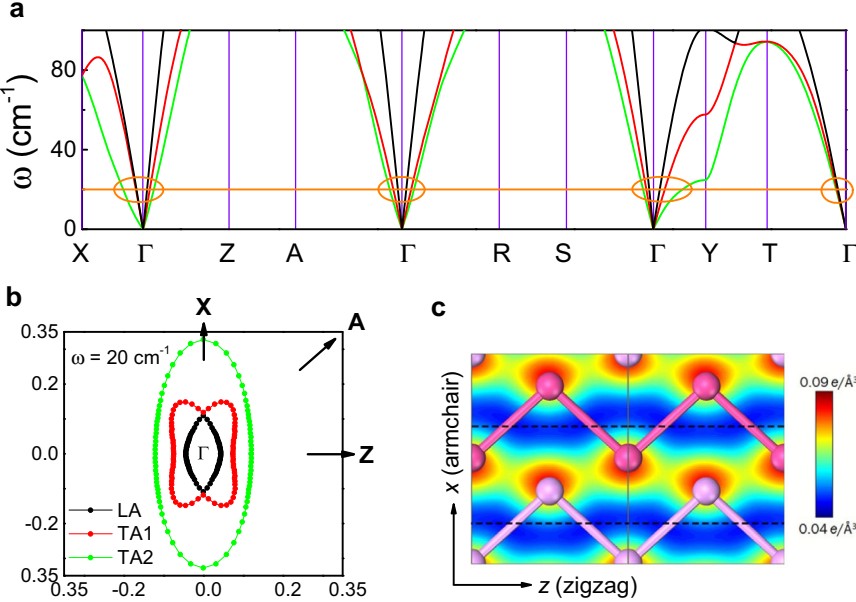

**Fig. 4 | Phonons in black phosphorus. a** Calculated phonon spectrum[34] below 100 cm⁻¹. The three acoustic branches are plotted with different colors. This spectrum, first calculated in ref. [34] is in agreement with a recent X-ray experimental study[48]. **b** Angle dependence of the phonon wave-vectors with a frequency of 20 cm⁻¹ projected in the $xz$ plane. The longitudinal mode (in black) and a transverse mode (in red) are almost degenerate along $x$. Energy transfer across phonon branches can be affected by the magnetic field. **c** Spatial distribution of charge concentration in bulk BP[35]. Note the x–z anisotropy. The (**c**) is adapted from ref. [35], Royal Society of Chemistry.

## Calculations

The phonon dispersions were obtained using the dynamical matrices calculated in ref. [34]. These calculations were performed using density functional perturbation theory[40] as implemented in the QUANTUM ESPRESSO package[41]. The Perdew, Burke, and Ernzerhof's generalized gradient approximation[42] and Garrity et al.'s pseudopotentials[43] were used. The van der Waals interaction was taken into account using Grimme's semiempirical recipe[44]. Planewave cutoffs of 50 and 250 Ry were used for the basis-set and charge density expansions, respectively. A $12 \times 12 \times 12$ $k$-point grid was used for the Brillouin zone integration in the self-consistent density functional theory calculations with a Marzari-Vanderbilt smearing of 0.02 Ry. The dynamical matrices were calculated on an $8 \times 8 \times 8$ $q$-point grid, and the phonon dispersions and density of states were obtained by Fourier interpolation.

## Data availability

The data that support the findings of this study are available from the corresponding author upon reasonable request.

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

## Acknowledgements

We thank Bruce Normand and Wei Ji for authorizing us to use Fig. 4c. This work was supported by The National Key Research and Development Program of China (Grant No.2022YFA1403503), the National Science Foundation of China (Grant No.12004123, 51861135104 and 11574097) and the Fundamental Research Funds for the Central Universities (Grant no. 2019kfyXMBZ071). Y.M. acknowledges funding from Grants-in-Aid for Scientific Research (Grant No. 19H01840). A.S. acknowledges computational resources provided by GENCI-TGCC (grant A0130913028). K.B. was supported by the Agence Nationale de la Recherche (ANR-19-CE30-0014-04). X.L. acknowledges the China National Postdoctoral Program for Innovative Talents (Grant No.BX20200143) and the China Postdoctoral Science Foundation (Grant No.2020M682386).

## Author contributions

X.L., Z.Z. and K.B. conceived of and designed the study. X.L. performed the transport measurements. A.S. performed the phonon spectrum calculations. X.L., Y.M., A.S., Z.Z., L.L., and K.B. analyzed the data. X.L., Z.Z. and K.B. wrote the paper with assistance from all the authors.

## Competing interests

The authors declare no competing interests.
