## [Peer Review File · Nature Communications]

The phonon thermal Hall angle in black phosphorusREVIEWER COMMENTS

Reviewer #1 (Remarks to the Author):

Large thermal Hall effect (THE) has recently been found in various quantum materials, such as underdoped cuprates, quantum spin liquid candidate α - RuCl_3 and quantum paraelectric SrTiO_3 . Although the THE has been thought to be a hallmark of nontrivial magnetic or fermionic excitations, accumulating studies suggest that phonons also play important roles in THE. X. Li and coworkers report large THE in black phosphorus (BP), which is a non-magnetic insulator. The authors found that the longitudinal and transverse thermal conductivities peak at the same temperature, which appear to be universal in various insulators displaying THE. Moreover, the thermal Hall angle shares similar magnitude among different insulators showing THE although the longitudinal thermal conductivity varies by orders of magnitude.

The results are very interesting and can provide important insights to the long debating mechanisms of THE. This article is well-written and proves high originality and importance of the study. Thus, I recommend the publication in Nature Communications.

However, before acceptance, few points need to be addressed:

1. The universal scaling between κ_{xx} and κ_{xy} is peculiar. As shown in Fig. 3, in various insulators, κ_{xx} and κ_{xy} not only peak at the same temperature, but also share similar temperature dependence. Could it be that the measured κ_{xy} is actually partial leakage of κ_{xx} ?

Measurements of the transverse temperature gradient are extremely challenge. How to make sure that the measured temperature gradient is coming from the sample? The longitudinal thermal gradient also shows magnetic response (Fig. S1), which is unusual for a non-magnetic insulator. Have the authors calibrated the thermocouples and thermometers in magnetic fields? Maybe one can present similar measurements on insulators which do not display THE.

2. The scaling ratio of κ_{xx}/κ_{xz} and κ_{zz}/κ_{zx} differs by one order of magnitude. Can the author discuss more on this anisotropy?

3. The authors speculated about possible origins of the THE in BP, i.e. anti-crossing of acoustic phonon branches and anisotropic charge distributions. But these features are unique to BP only and cannot explain the 'universal' THE behavior seen in various insulators.

4. The authors mentioned the diffuson-like channels in explaining glass-like thermal conductivity. Have the authors evaluated the contributions of diffusions in BP? The close packing of acoustic branches may give rise to large inter-branch coupling.

5. The estimation of thermal length is not straight forward. Can the authors elaborate more on that?

Reviewer #2 (Remarks to the Author):

Li et al. report measurements of magneto-thermal conductivities of black phosphorous (BP) films for wide ranges of temperature, magnetic field and sample thickness. The paper discovers a surprisingly large phonon-driven thermal Hall effect (THE) in BP.

The authors describe careful characterization of sample geometry, temperature probes, antisymmetrization with respect to magnetic field and subtraction of small electronic contributions to the thermal conductivity. Their systematic comparison of BP to other insulators exposes a very interesting observation: that the thermal Hall angle (ratio of thermal-Hall to thermal conductivity) seems to be of the same order of magnitude in insulators with widely different thermal conductivities, structures and magnetic and dielectric properties. This observation poses a serious challenge to the leading theoretical proposals which are currently on the table.

For a clean system one naively expects intrinsic mechanisms of the THE to yield a strong dependence of the thermal Hall angle on the phonon mean free path, This appears to be ruled out by the systematics shown in the paper. On the other hand, the high thermal conductivity of BP seems to rule out the importance of extrinsic scattering mechanisms which have been proposed in the literature for the other systems.

Therefore, this paper adds important information to the fascinating phenomenon of THE of insulators, which has been occupying the community in recent years. Their results strongly suggest that the underlying mechanism should be universal, yet theory must go beyond the existing paradigm of weakly scattered harmonic phonons.

The paper is definitely worthy of publication in Nature Communications. However, for additional benefit to the readers, I recommend that the authors consider including the following information.

1. What is known (experimentally) about the dispersions of the optical phonons of BP?
2. What is the dielectric constant tensor of BP?
3. Thermal Hall conductivity vs T, for the different thickness values.

Reply to reviewers

We appreciate that both reviewers devoted time to our manuscript.

Reviewer #1

Large thermal Hall effect (THE) has recently been found in various quantum materials, such as underdoped cuprates, quantum spin liquid candidate α -RuCl₃ and quantum paraelectric SrTiO₃. Although the THE has been thought to be a hallmark of nontrivial magnetic or fermionic excitations, accumulating studies suggest that phonons also play important roles in THE. X. Li and coworkers report large THE in black phosphorus (BP), which is a non-magnetic insulator. The authors found that the longitudinal and transverse thermal conductivities peak at the same temperature, which appear to be universal in various insulators displaying THE. Moreover, the thermal Hall angle shares similar magnitude among different insulators showing THE although the longitudinal thermal conductivity varies by orders of magnitude.

The results are very interesting and can provide important insights to the long debating mechanisms of THE. This article is well-written and proves high originality and importance of the study. Thus, I recommend the publication in Nature Communications.

Reply: We thank the referee for this accurate summary of our results and the recommendation for publication in Nature Communications.

However, before acceptance, few points need to be addressed:

1. The universal scaling between κ_{xx} and κ_{xy} is peculiar. As shown in Fig. 3, in various insulators, κ_{xx} and κ_{xy} not only peak at the same temperature, but also share similar temperature dependence. Could it be that the measured κ_{xy} is actually partial leakage of κ_{xx} ?

Reply: This is an important experimental point. However, there is fundamental difference between the two signals. The transverse response κ_{xy} is an odd function of the magnetic field: $\kappa_{xy}(B) = -\kappa_{xy}(-B)$, while the longitudinal response κ_{xx} is an even to field: $\kappa_{xx}(B) = \kappa_{xx}(-B)$. Therefore, the longitudinal leakage signal (due to an unavoidable misalignment between contacts) can be cancelled via the asymmetric processing of $\kappa_{xy} = (\kappa_{xy}(B) - \kappa_{xy}(-B))/2$. See the Section 2 of SM for more details.

Measurements of the transverse temperature gradient are extremely challenge. How to make sure that the measured temperature gradient is coming from the sample? The longitudinal thermal gradient also shows magnetic response (Fig. S1), which is unusual for a non-magnetic insulator. Have the authors calibrated the thermocouples and thermometers in magnetic fields? Maybe one can present similar measurements on insulators which do not display THE.

Reply: Yes, measuring the transverse temperature gradient is very challenging. We improved accuracy in two ways: i) Sweeping field from +12 T to -12 T multiple cycles at each temperature point, only overlapping data are considered reliable, as seen in Fig.S1. ii) At each field point and at each temperature, we repeated measurements many times to converge.

Fig-R1: The thermal Hall angle in PMN-PT

Indeed, as seen in Fig.S1, the longitudinal thermal gradient displays a very small (0.2% at 12 T) field dependence. This implies that longitudinal phonon thermal conductivity shows a very small field dependence in this non-magnetic. We note that a finite field dependence of thermal conductivity was previously reported in another non-magnetic insulator, that is InSb (See Jin et al., Nature Materials 14, 601 (2015)). Nevertheless, we cannot rule out that the longitudinal response is contaminated by possible field dependence of thermocouples' thermopower. However, this effect is even in magnetic field and does not affect the transverse signal, which is odd in magnetic field.

It is indeed a good idea to repeat the measurements on an insulator which do not display THE. Yes, we performed similar measurements on PMN-PT crystal using the same set-up and measurement system. This is an insulator with the chemical formula of $[\text{Pb}(\text{Mg}_{1/3}\text{Nb}_{2/3})\text{O}_3]_{(1-x)}[\text{PbTiO}_3]_x$ ($x=0.3$). We found a vanishing THE signal with $\kappa_{xy}/\kappa_{xx} \sim 0.01\%$, which is more than one order of magnitude smaller than the signal observed in BP (See the figure). This excludes that our set-up produces a finite THE because of an uncontrolled field dependence of the thermocouple thermopower.

2.The scaling ratio of κ_{xx}/κ_{xz} and κ_{zz}/κ_{zx} differs by one order of magnitude. Can the author discuss more on this anisotropy?

Reply: Thanks for this suggestion! We have added a few sentences to the main text for clarification: "It's worth noting that the κ_{ij}/κ_{ii} ratio is anisotropic. This is unavoidable and due to the combination of the Onsager reciprocity ($\kappa_{ij} = -\kappa_{ji}$) and the anisotropy of longitudinal thermal conductivity ($\kappa_{ii} \neq \kappa_{jj}$)".

3.The authors speculated about possible origins of the THE in BP, i.e. anti-crossing of acoustic phonon branches and anisotropic charge distributions. But these features are unique to BP only and cannot explain the 'universal' THE behavior seen in various

insulators.

Reply: Yes, the referee is perfectly right. What is specific to BP, is that the phonon spectrum is established in detail, and we can identify an anti-crossing between two phonon modes as a potential source of the observed in THE. In magnetic solids, the potential candidate can be another bosonic mode, possibly a magnon, yet to be identified in each material. The universality seen in the amplitude of the Hall angle and the correlation between the peaks in transverse and longitudinal responses does not imply that both players are identical. In our opinion, it strongly suggests that in all cases, one of the players is a heat-carrying acoustic phonon mode.

4. The authors mentioned the diffusion-like channels in explaining glass-like thermal conductivity. Have the authors evaluated the contributions of diffusions in BP? The close packing of acoustic branches may give rise to large inter-branch coupling.

Reply: Yes, indeed! The referee has a valid point. The phonon spectrum is full of branches close to each other and therefore there should be abundant interbranch coupling. However, we are unable to quantify the relative contribution of the diffusion-like channel to the total thermal conductivity in black P. We hope that our work stimulates future theoretical works exploring this.

5. The estimation of thermal length is not straight forward. Can the authors elaborate more on that?

Reply: Let us begin with the well-established case of electrons. In the semiclassical picture, the Hall angle is $\frac{\sigma_{xy}}{\sigma_{xx}}$. In two dimensions, this ratio is equal to $\frac{\sigma_{xy}}{\sigma_{xx}} = \frac{1}{\ell_B^2} \frac{\ell}{k_F}$.

Here, $\ell_B = \sqrt{\frac{\hbar}{e}}$ is the magnetic length. In other words, multiplying the Hall angle of electrons would yield a length, which is the geometric average of the mean-free-path and the Fermi wavelength. The same recipe can be used to extract a length from the thermal Hall angle of phonons. In many (but maybe not all) extrinsic scenarios, one would expect to see a correlation between the amplitude of the Hall angle and the mean-free-path of phonons. This is not what the experiment sees.

Reviewer #2 (Remarks to the Author):

Li et al. report measurements of magneto-thermal conductivities of black phosphorous (BP) films for wide ranges of temperature, magnetic field and sample thickness. The paper discovers a surprisingly large phonon-driven thermal Hall effect (THE) in BP.

The authors describe careful characterization of sample geometry, temperature probes, antisymmetrization with respect to magnetic field and subtraction of small electronic contributions to the thermal conductivity. Their systematic comparison of BP to other insulators exposes a very interesting observation: that the thermal Hall angle (ratio of thermal-Hall to thermal conductivity) seems to be of the same order of magnitude in insulators with widely different thermal conductivities, structures and magnetic and dielectric properties. This observation poses a serious challenge to the leading theoretical proposals which are currently on the table.

For a clean system one naively expects intrinsic mechanisms of the THE to yield a strong dependence of the thermal Hall angle on the phonon mean free path, This appears to be ruled out by the systematics shown in the paper. On the other hand, the high thermal conductivity of BP seems to rule out the importance of extrinsic scattering mechanisms which have been proposed in the literature for the other systems.

Therefore, this paper adds important information to the fascinating phenomenon of THE of insulators, which has been occupying the community in recent years. Their results strongly suggest that the underlying mechanism should be universal, yet theory must go beyond the existing paradigm of weakly scattered harmonic phonons.

The paper is definitely worthy of publication in Nature Communications. However, for additional benefit to the readers, I recommend that the authors consider including the following information.

Reply: We thank the reviewer for the recognition of our work and recommendation for publication.

1. What is known (experimentally) about the dispersions of the optical phonons of BP?

Reply: Recently, Pogna and collaborators studied the lattice dynamics of black phosphorus using a combination of x-ray diffuse scattering and inelastic x-ray scattering (Pogna et al, Phy. Rev. B 105, 184306 (2022)). They also performed *ab initio* calculations, which yielded a phonon spectrum compatible with their experimental results. As seen in the figure below, their results is close to the theoretically calculated phonon spectrum used in our study and first reported in Machida *et al.* , Sci. Adv. 4: eaat3374 (2018). At this stage, there is no evidence for a soft optical phonon with temperature-dependent dispersion. Being an elemental solid, BP cannot become ferroelectric. However, let us note that Black P and SnSe are isostructural, and ferroelectricity has been discovered in single-layer SnSe (K. Chang et al, Nano Lett. 20, 6590–6597(2020)).

Fig-R2: Phonon spectrum in Black phosphorus according to Pogna et al, *Phy. Rev. B* 105, 184306 (2022) (top) and Machida et al. , *Sci. Adv.* 4: eaat3374 (2018) (bottom). Note similarity of dispersions along Γ -Z, Γ -Y and Γ -S orientations.

2. What is the dielectric constant tensor of BP?

Reply: Nagahama and co-workers (*J. Phys. Soc. Jpn.* 54 2096 (1985) have measured the experimental dielectric constant of BP along different orientations. Valagiannopoulos *et al.* (*J. Phys. Commun.* 1 045003 (2017)) have calculated the real and imaginary parts of the dielectric constant. The real components of their calculated values are in good agreement with what was measured and show a modest anisotropy. Interestingly, there is a huge anisotropy in the imaginary component. As a consequence, an electromagnetic wave is damped much less along the ‘zigzag’ direction than along the ‘armchair direction. We have referred to this feature in the new version and added a new section on this point in the supplement.

3. Thermal Hall conductivity vs T , for the different thickness values.

Reply: We did not study the thickness dependence of the thermal Hall effect. In the three BP samples subject to our THE investigation, the thickness varied from 30 to 70 microns. We did not detect any significant evolution with thickness. This emerges as a subject for future studies.

List of changes:

1. All changes in text are marked in red.
2. Three references (38, 42 and 50) are added.

REVIEWERS' COMMENTS

Reviewer #1 (Remarks to the Author):

The authors have addressed all of my concerns. The revised manuscript is suitable for the publication in Nature Communications. The universal thermal Hall angle seen in various insulators certainly provides important clues to the origins of THE. This work can further motivate more experimental and theoretical studies on THE.

A minor point:

In the discussion part, I would recommend the authors elaborate more on the common features and differences among the presented insulators. Adding the following part of the response letter to the manuscript could be sufficient.

'In magnetic solids, the potential candidate can be another bosonic mode, possibly a magnon, yet to be identified in each material. The universality seen in the amplitude of the Hall angle and the correlation between the peaks in transverse and longitudinal responses does not imply that both players are identical. In our opinion, it strongly suggests that in all cases, one of the players is a heat-carrying acoustic phonon mode.'

Reviewer #2 (Remarks to the Author):

The authors have answered the questions raised by both referees adequately, and added the necessary clarifications in the revised manuscript. I recommend accepting the current version for publication in Nature Communications.

Reply to reviewers

We appreciate that both reviewers devoted time to our manuscript.

Reviewer #1 (Remarks to the Author):

The authors have addressed all of my concerns. The revised manuscript is suitable for the publication in Nature Communications. The universal thermal Hall angle seen in various insulators certainly provides important clues to the origins of THE. This work can further motivate more experimental and theoretical studies on THE.

We thank the reviewer for the recognition of our work and recommendation for publication.

A minor point:

In the discussion part, I would recommend the authors elaborate more on the common features and differences among the presented insulators. Adding the following part of the response letter to the manuscript could be sufficient.

'In magnetic solids, the potential candidate can be another bosonic mode, possibly a magnon, yet to be identified in each material. The universality seen in the amplitude of the Hall angle and the correlation between the peaks in transverse and longitudinal responses does not imply that both players are identical. In our opinion, it strongly suggests that in all cases, one of the players is a heat-carrying acoustic phonon mode.'

Thanks for this good suggestion! We have added the following paragraph to the revised manuscript:

We saw that the transverse and longitudinal conductivities peak at the same temperature in a variety of insulators. This strongly suggests a prominent role played by heat-carrying acoustic phonon in generating the THE even in magnetic materials. However, this does not exclude a role played by magnetism. In magnetic materials, a phonon can couple to another bosonic mode (and possibly a magnon) specific to each material.

Reviewer #2 (Remarks to the Author):

The authors have answered the questions raised by both referees adequately, and added the necessary clarifications in the revised manuscript. I recommend accepting the current version for publication in Nature Communications.

We thank the reviewer for the recommendation.

Changes:

1. A paragraph mentioned above has been added.
2. The section titles have been added.